# Exosomes as Intercellular Messengers in Hypertension

**DOI:** 10.3390/ijms222111685

**Published:** 2021-10-28

**Authors:** Olufunke Omolola Arishe, Fernanda Priviero, Stephanie A. Wilczynski, R. Clinton Webb

**Affiliations:** 1Cardiovascular Translational Research Center, University of South Carolina, Columbia, SC 29209, USA; fernanda.priviero@uscmed.sc.edu (F.P.); Stephanie.Wilczynski@uscmed.sc.edu (S.A.W.); clinton.webb@uscmed.sc.edu (R.C.W.); 2Department of Cell Biology and Anatomy, University of South Carolina, School of Medicine, Columbia, SC 29209, USA

**Keywords:** hypertension, exosomes, intercellular communication

## Abstract

People living with hypertension have a higher risk of developing heart diseases, and hypertension remains a top cause of mortality. In hypertension, some detrimental changes occur in the arterial wall, which include physiological and biochemical changes. Furthermore, this disease is characterized by turbulent blood flow, increased fluid shear stress, remodeling of the blood vessels, and endothelial dysfunction. As a complex disease, hypertension is thought to be caused by an array of factors, its etiology consisting of both environmental and genetic factors. The Mosaic Theory of hypertension states that many factors, including genetics, environment, adaptive, neural, mechanical, and hormonal perturbations are intertwined, leading to increases in blood pressure. Long-term efforts by several investigators have provided invaluable insight into the physiological mechanisms responsible for the pathogenesis of hypertension, and these include increased activity of the sympathetic nervous system, overactivation of the renin–angiotensin–aldosterone system (RAAS), dysfunction of the vascular endothelium, impaired platelet function, thrombogenesis, vascular smooth muscle and cardiac hypertrophy, and altered angiogenesis. Exosomes are extracellular vesicles released by all cells and carry nucleic acids, proteins, lipids, and metabolites into the extracellular environment. They play a role in intercellular communication and are involved in the pathophysiology of diseases. Since the discovery of exosomes in the 1980s, numerous studies have been carried out to understand the biogenesis, composition, and function of exosomes. In this review, we will discuss the role of exosomes as intercellular messengers in hypertension.

## 1. Introduction

Despite impacting nearly 1.13 billion people worldwide [1], the pathophysiology of hypertension remains elusive. It is a condition that can lead to heart disease, vascular damage, renal injury, and other detrimental conditions that can impact a patient’s quality of life and overall health. Globally, hypertension is responsible for approximately 7.6 million deaths per year, making the race for the discovery of new markers, pathways, and causes of this disease much more crucial [2]. In a healthy individual, blood pressure (BP) can vary from minute to minute, with normal values corresponding to <120 mmHg systolic blood pressure (SBP)/<80 mmHg diastolic blood pressure (DBP)^3^. As arterial blood pressure rises, a patient is at risk for developing hypertension and other vascular diseases. Obesity, aging, and decreased physical activity are some of the key risk factors that are seen before the emergence of hypertension, although genetic factors and environmental conditions can also influence a patient’s predisposition or susceptibility to hypertension.

Elevated arterial blood pressure in hypertension is attributable to increased total peripheral resistance, which results, at least in part, from alterations in humoral and neurogenic components and vascular endothelial and smooth muscle dysfunction [3].

Indeed, in the last two decades, much effort was made to address the mechanisms involved in the development of hypertension. Increased peripheral resistance is largely attributed to vascular remodeling (either inward eutrophic remodeling or hypertrophic remodeling) and endothelial dysfunction, and these modifications are consequences of an exacerbation in physiological processes such as apoptosis, inflammation, and vascular fibrosis [4,5]. In this context, exosomes become a potential source of molecules that could trigger these processes, since exosomes are released from the cells through exocytosis, and the content of exosomes can vary from cell signaling molecules to cell waste; further, exosomes can contain secretions meant to enhance the cell’s local environment [6]. In addition, exosomes can also contain genetic material from the cell, which can act on its target and can modulate the cell’s activity [7]. Therefore, the involvement of exosomes may be crucial to understanding the systemic communication of cells in hypertensive patients and can also shed light on the advancement of the pathology of hypertension.

## 2. Exosomes

Composed of a lipid bilayer containing transmembrane proteins and enclosed cytosolic proteins and RNA, exosomes were first described in 1981 in a culture of normal or neoplastic cells that were able to exfoliate micro-vesicles of 500–1000 nm diameter, containing a second population of vesicles of about 40 nm diameter, with the ecto-enzyme activity of 5′-nucleotidase. The authors proposed the term exosomes for these micro-vesicles released from the plasma membrane [8]. Currently, exosomes are classified by most studies as extracellular vesicles measuring between 30 and 150 nm diameter that are formed by late endosomes [9]. The formation of exosomes will occur when the lumen of endosomes becomes full of intraluminal vesicles, leading to an inward budding and forming small vesicles containing endosome-derived molecules (multivesicular bodies or MVB) [10]. The small vesicles fuse to the plasma membrane to be secreted into the extracellular space through exocytosis (Figure 1), which suggested the term exosomes, and are known to promote cell-to-cell communication [11]. The selectivity of the cell-to-cell communication is given by the composition of the proteins/glycoproteins present on the surface of exosomes and target cells [12]. Exosomes uptake will occur by different mechanisms including endocytosis, micropinocytosis, phagocytosis, and internalization [13]. Functionally, several activities have been ascribed to exosomes. For example, during the maturation of reticulocytes, the exosomes present a variety of activities, including acetylcholinesterase, cytochalasin B binding, nucleotide binding, amino acid transport, and transferrin receptor expression. It is believed that the formation of these micro-vesicles is a mechanism to shed specific functions of the membrane, which is necessary only during the maturation of reticulocytes and will decrease with the formation of the mature erythrocytes [14]. Other functional activities for exosomes have been further described. In B cells, the membrane of major histocompatibility complex (MHC) class II-enriched compartments (MIIC) was found to fuse with the plasma membrane to promote the exocytosis of exosomes carrying MHC class II molecules and introducing a possible mechanism of transport for the exosomes [15]. It was suggested that these exosomes could be transferring units of MHC class II molecules between different cells of the immune system [15], since MHC class II molecules are known for presenting antigenic peptides to CD4+ T cells [16]. Interestingly, the literature has demonstrated a role for exosomes in physiological and pathological conditions, such as in the progression of cancer, which has been vastly described in the literature [17,18,19,20]. In addition, circulating and urinary exosomes are being used to identify key proteins and microRNA (miRNAs), circular RNAs (circRNA), or long non-coding RNA (lncRNA) associated with different diseases including type 2 diabetes, nephropathy, aldosteronism, atherosclerosis, and several types of cancer (bladder, gastric, prostate cancers) [21,22,23,24,25]. Therefore, it seems that the exosomes carry potential molecules to help in the diagnosis of diseases. On the other hand, the use of exosomes as a potential therapy also seems to be promising, since exosomes exhibit some advantages including carrying natural molecules, displaying a natural targeting capacity, and others, such as the delivery of miRNA or other contents directly to the cytoplasm, bypassing the endosomal pathway and lysosomal degradation, which might decrease the susceptibility to degradation/modification of its content [26,27,28,29,30]. This can be explained because exosomes may deliver their content through three distinct mechanisms, including (1) interaction with adhesion molecules or receptors present on target cells, resulting in the initiation of downstream signaling; (2) a direct release of their content after a direct fusion with the plasma membrane of the target cell; (3) after endocytosis, fusion with the endosomal membrane, releasing their content into the cytoplasm [31]. A recent review summarizes the state of art of the use of exosomes as biomarkers and drug delivery tools [32]. Despite the possible therapeutic role of exosomes, this review will focus on the content of exosomes and how it could contribute to the development of hypertension. 

### The Role of Exosomes as Mediators of Intercellular Communication

Cells communicate in different ways, i.e., by direct contact and through paracrine, endocrine, and synaptic transmission [33]. The exosomes play a role in intercellular communication and are involved in the pathophysiology of diseases [34]. The exosomes also contain proteins that are specific to the function of their originating cell, for example, exosomes from human urine contain aquaporin-2 derived from renal tubular epithelial cells [35]. The constituents of exosomes are so diverse that they contain proteins that mediate the functions of almost all the systems in the body, from proteins that are mediators of cell–cell communication, which include cytokines [36], hormones [37], growth factors [38], and heat shock proteins [39], to cell surface proteins [33]. Studies have shown that exosomes when injected are efficient at going into other cells and when they are in these target cells, they release their components which are biologically active [40,41,42]. Importantly, cargos within an exosome reflect the pathophysiological state of the originating cell [43]. The cargos released from exosomes are either taken up by the cells or can activate cell surface receptors. This leads to changes in the cellular phenotype.

The exosomes contain both mRNA and miRNA [44]. It has also been reported that the RNA from mast cell exosomes can be transferred to other mast cells and these transferred RNAs can be translated and are functional in their new (recipient) cell. This was observed when new mouse-specific proteins were found in the recipient cells after the transfer of exosomal RNA from mouse to human mast cells [44]. Because of their important role in intercellular communication, exosomes are important for the regulation of physiological processes in health and diseases [45].

## 3. The Role of Exosomes in Hypertension

Currently, the release of exosomes and their mediated signaling is attracting attention in the pathophysiology of diseases. In this section, we will discuss how exosomes play a role in the communication between cells under hypertensive conditions, pointing out how they contribute to the pathophysiology of the disease.

### 3.1. Importance of Exosomal miRNA in Hypertension

MicroRNA (miRNA) are small 18-to-28-nucleotide non-coding RNA molecules. They regulate post-transcriptional protein expression in healthy and pathological cellular processes. miRNA are synthesized from their precursor RNA (pre-miRs) by sequential cleavage by ribonuclease III enzymes—Dicer and Drosha—into mature miRNAs, which are then loaded to Argonaut to form the miRNA-induced silencing complex (miRISC) [46]. As early as 1972, it was thought that miRNA was freely circulating in the plasma; however, it was later discovered in 2008 that it is present in all other components of the circulation including red blood cells, platelets, and white blood cells [47]. miRNA are transported in the circulation in exosomes, where they are protected from degradation by the exosomal bilayer membrane [48]. These exosomal miRNAs are highly stable even when exposed to unfavorable conditions, unlike exogenous miRNAs which are quickly degraded by the activity of RNases in the plasma [49]. They are also delivered to target cells through exosomal transport, where they mediate gene translation [50]. This feature of miRNA has given rise to the idea that they might be a suitable biomarker of cardiovascular diseases, particularly, hypertension. This is because they fulfill the criterion of being an ideal biomarker [51]. Furthermore, exosomal miRNAs mirror the state of the originating cell, are functional in the target recipient cell, and their regulation is tissue-specific [44]. miRNAs are very important regulators of a wide range of biological processes including cellular proliferation, apoptosis, and differentiation. Therefore, any change in their expression might lead to cellular dysfunction, resulting in the development of disease. Generally, miRNAs regulate cellular pathways by regulating gene transcription. Specifically, miRNAs implicated in hypertension include endothelial miRNAs, renal miRNAs, miRNAs targeting vascular smooth muscle cells, and miRNAs targeting the renin–angiotensin–aldosterone system (RAAS) [52]. The role of miRNA in the pathophysiology of hypertension involves miRNA-mediated modulation of RAAS, nitric oxide (NO), oxidative stress, vascular inflammation, and angiogenesis [53]. Using next-generation sequencing, the miRNA expression profiles of plasma exosomes in spontaneously hypertensive rats (SHR) and Wistar Kyoto rats (WKY) were defined by a group who reported that the plasma exosomal miRNA expression of WKY differed significantly from that of SHR. Out of the 27 miRNAs that were reported to be expressed differently, 23 miRNAs were upregulated in exosomes from SHR compared with exosomes from normotensive WKY rats, and 10 out of the 23 miRNAs that were upregulated were confirmed to be involved in signaling pathways and genes that are specific for hypertension [54]. Exosomal miRNAs have been implicated in the development and consequences of hypertension. They play a role in the development of hypertension by negatively regulating transcript levels by binding to the 3′-untranslated region of mRNAs and destabilizing it [55]. It has been suggested that exosomes have an impact on blood pressure regulation through miR-425-5p because the expression of this miRNA was 2.74-fold greater in the exosomes of SHR than in those of WKY rats [54]. In a transcriptome-wide study of differential expression of mRNAs and miRNAs in the kidneys of hypertensive patients, the overexpression of a novel gene, *ren*, upregulated by neurogenic signals (retinoic acid, epidermal growth factor (EGF), and nerve growth factor (NGF)), was reported, coupled with the downregulation of two miRNAs that bind to *ren*, implicating renin. The authors also reported the role of other miRNAs implicated in the etiology of hypertension [56]. The specific miRNAs implicated in hypertension include miR-15 that targets hypertension-associated single-nucleotide polymorphisms (SNPs) in the angiotensin II type I receptor 3′UTR [55,57]. These polymorphisms alter L-arginine metabolism by altering the expression of type 1 cationic amino acid transporter SLC7A1, leading to reduced bioavailability of nitric oxide and endothelial dysfunction in hypertension [58,59]. Other exosomal miRNAs that could affect the pathophysiology of hypertension include miR-128-3p, whose expression was 3.06-fold higher in exosomes from SHR than in exosomes of WKY rats, and miR-128, whose expression was reported to be greater in hypertensive patients as compared to controls [60]. miR-128 regulates Kruppel-like factor 4 (KLF4), altering the proliferation, migration, differentiation, and contractility of vascular smooth muscle cells. miR-128 also methylates the VSMC gene myosin heavy chain 11 (Myh11) [61] through the induction of KLF4 regulation [61].

A key feature of hypertension is renal damage, and exosomes have been shown to play a role in this phenomenon. A study observed that in hypertensive patients with albuminuria, the profile of 29 plasma exosomal miRNAs was changed compared to controls [62]. These miRNAs play significant roles in the development of renal damage in hypertension.

Currently, the molecular etiology of hypertension has not been fully elucidated. The study of the role and the mechanisms through which exosomal miRNAs contribute to the development of hypertension will go a long way, possibly allowing the development of novel therapeutic targets for hypertension. For a detailed review of the role and mechanisms of miRNA in hypertension, you can see a recent publication [52].

Next, we will discuss the contribution of exosomes to mechanisms of vascular remodeling, endothelial dysfunction, and inflammation that are important contributors to the pathology of the hypertension.

### 3.2. Exosomal Regulation of Vascular Remodeling in Hypertension

It has been suggested that one of the earliest indications of target organ damage in hypertension is the remodeling of small vessels [63]. An increase in blood pressure stimulates vascular remodeling directly by stimulating mechanoreceptors and by increasing media stress [64].

Vascular remodeling, which is a hallmark of primary hypertension, leads to decreased arterial compliance and increased stiffness. Mechanisms of vascular remodeling in hypertension include the action of vasoactive peptides (Ang II and Endothelin-1), inducing vasoconstriction, vascular smooth muscle cell growth, and apoptosis, leading to inflammation and vascular fibrosis [4]. Every step of this mechanism requires intercellular communication. Therefore, an important process of vascular remodeling is intercellular communication. This was believed to be mediated by direct cell–cell communication or through paracrine mechanisms, but the importance of exosomes in intercellular communication in the process of vascular remodeling has been highlighted [65]. 

It was reported that plasma-derived exosomes from an animal model of hypertension (SHR) stimulate the medial wall thickness of the aorta in the artery of normotensive animals (WKY rats) [66]. The role of exosomal stimulation in arterial remodeling was elucidated in pulmonary hypertension. Translationally controlled tumor protein (TCTP), which plays an important role in pulmonary arterial vascular remodeling [67], was transferred from endothelial cells to pulmonary artery smooth muscle cells via exosomes [68]. TCTP was first observed to be greatly expressed in exosomes compared to microparticles, indicating that it is mainly transported via exosomes.

Ang II induces vascular inflammation by modulating vascular wall effectors by Ang II type 1 receptors (AT_1_Rs) [3]. Activation of these AT_1_Rs stimulates angiogenesis and inflammation of the vasculature. These processes interact with vasoconstriction to enhance remodeling.

In a study by Pironti et al., exosomes from AT_1_R-overexpressing cells exposed to osmotic stretch were isolated. Exosomes were also isolated from the sera of mice subjected to cardiac pressure overload. These exosomes were found to contain AT_1_Rs, and this was confirmed by electron microscopy, radiology, and receptor binding assays [69]. The authors also reported that these exosomes delivered the functional AT_1_Rs they contained to cells that were subjected to osmotic stretch (in vitro model of cellular stretch). Another group reported that exosomes from an animal model of hypertension (rats infused with Ang II) upregulated the expression of inflammatory factors in cultured human coronary artery endothelial cells, leading to endothelial dysfunction in these cells [70].

### 3.3. Exosomal Stimulation of Endothelial Dysfunction in Hypertension

The endothelium is composed of specialized cells located on the internal lining of the vasculature. The vascular endothelium is now regarded as a multifunctional organ [71]. It responds to signals including shear stress, hypoxia, and inflammation and releases factors that attune the vascular tone. The endothelium releases vasorelaxants, which include NO and endothelium-derived hyperpolarizing factors. Apart from these, endothelin-1, angiotensin II, superoxide ions, and thromboxane [72] are vasoconstrictive factors that are released by the endothelium. In physiological situations, the vasoconstrictive and vasorelaxants released by the endothelial cells are balanced, but in hypertension, this balance is altered, and this results in decreased vasodilation [73].

Endothelial cells secrete exosomes, which could regulate their function and integrity. The evidence for this was provided by a study that reported that the endothelial secretion of heat shock protein 70 (HSP70) is exosome-dependent and is important for the regulation of vascular endothelial integrity [74]. Furthermore, endothelial cells, being the barrier between the vasculature wall and the blood, can be targeted by exosomes released from different cells into the circulation. Depending on the constituents of the exosomes, the function of the targeted endothelium can be altered.

Exosomes contribute to the pathophysiology of hypertension by stimulating endothelial dysfunction. Exosomes can stimulate endothelial dysfunction by increasing endothelial tight junctions’ permeability and adhesion molecule expression. This was observed in human dermal microvascular endothelial cells (hMVEC-d) monolayers [75]. A group of hypertensive patients with obstructive sleep apnea (OSA) was studied, and their exosomes were isolated. Exosomes from reverse-dipping blood pressure (RDBP) patients caused an increase in the permeability of endothelial tight junctions and adhesion molecule expression, stimulating endothelial dysfunction in human dermal microvascular endothelial cells (hMVEC-d) monolayers [75]. Exosomes derived from serum under hypertensive conditions can also directly activate pro-inflammatory pathways in endothelial cells, leading to endothelial dysfunction [70].

### 3.4. Exosomal Stimulation of Inflammation in Hypertension

Inflammation arises when homeostasis between host and immune cells is lost [76]. Inflammation is a complex phenomenon, comprises the first response of the immune system in the presence of harmful stimuli, and is considered to be a major cause of increased blood pressure [77]. Human studies and studies with experimental models of hypertension have been carried out to understand the role of the immune system in the etiology of hypertension. Both arms of the immune system, the innate and the adaptive, coordinate the process of inflammation against pathogens or irritants. While human studies reported an independent association between markers of inflammation and hypertension [78,79,80], the role of the immune system in the development of hypertension has been well demonstrated in various models of hypertension [81,82,83,84]. The inflammatory response is mediated by complex interactions between inflammatory cells and vascular cells, leading to the increased expression of cytokines, chemokines, and growth factors [85]. Vascular inflammatory responses are mediated majorly by the I-κB/nuclear factor-κB (NFκB) system that is activated by stimuli including cytokines, protein kinase C activators, and reactive oxygen species [86]. After activation, NFκB translocates into the nucleus where it regulates the transcription of genes involved in the pathogenesis of inflammatory lesions. Indeed, these genes encode cytokines such as interleukin-6 and tumor necrosis factor-α, chemokines such as monocyte chemotactic protein 1 (MCP-1), and adhesion molecules such as intercellular adhesion molecule 1 (ICAM-1), vascular cell adhesion molecule 1 (VCAM-1), and platelet endothelial cell adhesion molecule, all involved in the recruitment of monocytes/macrophages to sites of inflammation in the vascular wall [87].

Exosomes have been implicated in the etiology of inflammatory diseases including cancer and diabetes. Indeed, exosomes’ involvement in immunoregulatory mechanisms includes activation or suppression of the immune response, modulation of antigen presentation, and immune surveillance [88]. Exosomal cargoes include DNA and miRNA which could activate the innate and adaptive immune responses [89]. They activate the immune response by transferring miRNAs to the recipient cells, and these miRNAs stimulate gene signaling and dendritic cell maturation [90]. In cancer, tumor-derived exosomes activate NF-κB, leading to the activation and release of pro-inflammatory cytokines in macrophages. Exosomes also deliver miRNA to cancerous cells to stimulate tumor-inhibiting effects [91,92,93,94,95,96]. It was reported that the expression of plasma exosomal miR-15a was higher in diabetic patients compared to controls. miR-15a contributes to disease severity by increasing oxidative stress in the patients [97,98]. Although a few studies have been performed to examine the role of exosomal pro-inflammatory factors in the progression of the disease, the specific role of exosomes in the inflammatory process of hypertension has not been fully elucidated. It was reported that the levels of the pro-inflammatory factors intracellular adhesion molecule-1 (ICAM1) and plasminogen activator inhibitor-1 (PAI-1) were increased in the blood vessels of Ang II-induced hypertensive rats [70]. Another study reported that the exosomal miRNA that is linked to inflammation, miR-17-5p, was increased in hypertensive animals compared to normotensive controls [54]. More studies need to be performed to fully understand the mechanisms of exosomal-derived factors in the etiology of hypertension. 

### 3.5. Exosomes in the Development of Preeclampsia

Preeclampsia is an ailment that manifests in the second half of pregnancy. It is associated with inadequate placental formation, chronic inflammation, maternal vascular dysfunction, hypertension, proteinuria, and organ dysfunction in the pregnant woman, and a decline in fetal growth [99]. It affects approximately 3–10% of all pregnancies and exposes the maternal system to complications that could lead to fetal and maternal mortality in pregnancy [100]. It is associated with an increased risk of developing cardiovascular disease later in maternal life. Preeclampsia is a multisystem disease, and it is still unknown what induces it, though it is believed to be rooted in shallow or inadequate placentation [101]. However, the exact mechanism responsible for insufficient placentation and how it leads to the development of increased blood pressure and cardiovascular disease remain unclear. Currently, the only effective treatment is the delivery of the fetus and the placenta. In addition to the cardiovascular risks to the mother and fetus during pregnancy, preeclampsia is independently associated with a higher risk of cardiovascular disease later in maternal life [102,103,104]. 

Studies have demonstrated that the concentration of peripheral blood exosomes is significantly increased in hypertensive pregnant women compared with normal pregnancies and directly correlates with disease severity [105,106]. Thus, exosomes are currently being evaluated as potential biomarkers of preeclampsia. Changes in the release, concentration, composition, and bioactivity of exosomes in preeclampsia versus normotensive pregnancy have also been reported. During normal pregnancy, the concentration of exosomes is increased as the pregnancy progresses; this increase is greater in preeclampsia than in normal pregnancy [105,107]. Because exosomes can be isolated from the blood of women in early pregnancy, they can be targeted in the intervention for preeclampsia and also in the early prediction of the development of the disease. Studies have shown that most maternal exosomes are derived from a variety of maternal cells including B cells, T cells neutrophils, and endothelial cells [108,109]. In addition to these constituent cells, the placenta also releases exosomes into the maternal circulation [110]. The concentration of exosomes originating from the placenta increases in a time-dependent manner throughout pregnancy [111]. Exosomes are taken up into their target maternal cells by endocytosis. Endoglin, which plays an important role in the regulation of the vascular structure, is a supplementary receptor for transforming growth factor β [112]. It is carried in exosomes in the circulation, and the circulating levels of exosomal endoglin are increased in preeclamptic women [113]. Furthermore, a study reported that exosomes derived from preeclamptic women expressed abundant circulating soluble fms-like tyrosine kinase-1 (sFlt-1) and soluble endoglin (sEng). sEng is considered to be a marker of preeclampsia, because its circulating levels are increased in patients with this disease [114]. An elevated blood pressure in pregnant mice that were given exosomes from preeclamptic women was also observed, and these mice also had a decreased body weight compared to the controls. The embryos from these mice had lower birth weights, and there was a decrease in the number of surviving embryos per litter [115]. 

## 4. Targeting Exosomes in Hypertension

Generally, it has been established that the exosomal content of miRNAs increases in pathological states, for example, in pulmonary hypertension [116], cancer [117], and preeclampsia [118]. In hypertension, there is an exacerbation of some physiological processes including apoptosis, inflammation, and vascular fibrosis. This leads to vascular remodeling and endothelial dysfunction, which are the major mechanisms that contribute to the pathophysiology of the disease [119]. As highlighted in this review, exosomes have been shown to contribute to these mechanisms. In addition, exosomes, depending on their donor cells, express some particular lipids and proteins including cell adhesion molecules and ligands [120]. These lipids and proteins expressed by exosomes could be used to identify and isolate their cells of origin, thereby enabling us to target the particular tissue or region that could be undergoing apoptosis, inflammation, or vascular fibrosis as opposed to just targeting a whole organ system blindly. This knowledge could also help researchers quantify each particular cell-derived exosome to understand which particular set of exosomes is being upregulated or downregulated and any alteration in their cargos depending on the state of the organism.

Furthermore, exosomal contents are not only altered in hypertension but the contents of the exosomes in hypertension are bioactive and could stimulate dysfunction in the target cells. Therefore, studying the exosomal content in hypertension could lead to the development of potential targets for the diagnosis, prevention, and treatment of this disease.

Exosomes contribute to the pathophysiology of diseases by delivering their content to target cells, thereby increasing inflammation, cell proliferation, and migration [121,122]. When exogenous exosomes are injected, they enter the cell in the organism that was injected and release their cargo which is functional and protected from immune clearance [123]. Importantly, the cargos within an exosome reflect the pathophysiological state of the originating cell [43]. The cargos can activate cell surface receptors or/and be taken up by or incorporated into recipient cells, leading to changes in cellular phenotype. They have been studied for their potential as non-invasive prognostic and diagnostic biomarkers [124] and, more recently, as therapeutic nano carriers [125]. 

### 4.1. Exosomes as Biomarkers in Hypertension

The importance of exosomes in the etiology of diseases has been documented. One unique feature of exosomes is that they are found and can be isolated in all biological fluids including blood [89], urine [126], saliva [127], breastmilk [128], cerebrospinal fluid [129], semen [130], amniotic fluid [131], nasal fluid [132], and ascites [133]. Furthermore, unhealthy or diseased cells secrete more exosomes than healthy cells [134]. These features make them a suitable candidate as a minimally invasive biomarker for the diagnosis or study of the progression of diseases including cardiovascular diseases [135,136], cancer [137,138], and other diseases [139,140,141,142]. They can also be useful for studying physiological processes; for example, salivary exosomal miRNAs were identified as suitable biomarkers for aging [127]. 

Apart from this feature, the exosomal cargo is complex and consists of lipids, proteins, mRNA, miRNA, and DNA that could be targeted and studied to understand the underlying pathophysiological mechanisms of diseases [143]. For example, the proteomic content of exosomes from human cancerous pleural effusions was studied to identify the specific proteins that were altered and that could contribute to the pathogenesis of the disease [144]. 

The potential of exosomes as markers of hypertension has been highlighted [145]. Hypertension is a complex, multifactorial disease, and its development is determined by a combination of genetic susceptibility and environmental factors [146]. Long-term efforts by several investigators have provided invaluable insight into the physiological mechanisms responsible for the pathogenesis of hypertension [147,148,149,150]. Exosomes, being responsible for intercellular communication, could play significant roles in modulating some of these mechanisms, thereby representing a suitable research target to understand the etiology and progression of the disease. In hypertension, blood-derived exosomes contain DNA, miRNA, mRNA, and other proteins that can alter the functional state of their target cells (Figure 2). Any alteration in the cargo of exosomes could also provide information that would help us understand the underlying molecular mechanisms of the disease [151]. 

### 4.2. Exosomes as Potential Therapeutic Agents in Hypertension

Exosomes are being considered as therapeutic targets given their ability to transport proteins and nucleic acids preventing their rapid degradation, their low cytotoxicity in the circulation, and their low biohazardous potential [134]. Apart from this, exosomes uniquely have high biocompatibility and also contain specific proteins that help them to identify and target their destination cells [31]. 

Studies have shown that stem cell-derived exosomes could be useful in the treatment of pulmonary hypertension. Evidence for this includes a study that showed that in pulmonary hypertension, treating hypertensive rats with Mesenchymal Stromal Cell-Derived Exosomes (MSC-EXO) improved the phenotype [152,153]. Another study also showed that MSC-EXO, when injected, attenuates pulmonary vascular remodeling and right ventricular damage in monocrotaline-induced hypertensive rats. It also inhibited vascular endothelial cell apoptosis and smooth muscle cell proliferation [154]. MSC-EXO also improved lung function and attenuated inflammation in mice exposed to hyperoxia [155]. Moreover, stem cell-derived exosomes attenuated mitochondrial dysfunction in an animal model of pulmonary hypertension [156].

## 5. Conclusions and Future Directions

The role of exosomes in the pathophysiology of diseases is extensive, and a single review cannot exhaustively summarize the current knowledge available. In hypertension, the current knowledge indicates that exosomes play a role in the pathophysiology of the disease. Exosomes derived from hypertensive patients can be studied to reveal the state of the parents’ cells, because the exosomal content reflects the state of the parents’ cells. Alterations in miRNA, DNA, and other contents of exosomes could help understand the molecular mechanisms underlying the pathophysiology of hypertension. This knowledge could go a long way allowing the development of therapeutic interventions specific to individuals, as specific exosomal proteins can be targeted by either upregulating or downregulating them.

As highlighted in this review, exosomes, through their contents, modulate vascular and endothelial function in hypertension. Therefore, future studies should focus on elucidating the various content of exosomes in hypertensive patients, particularly highlight the components that are altered compared to normotensives individuals. Furthermore, the mechanisms through which each component of exosomal content individually or collectively exacerbates the disease process should also be studied. This will help to better understand and the use of exosomes as therapeutic targets in health and disease.

Currently, exosomes are easily isolated from body fluids, and this makes them accessible for isolation and study. While techniques for the isolation of exosomes, including ultracentrifugation, ultrafiltration, and a charge-based precipitation method involving protamine sulfate [157], are readily available, there are limited techniques available to identify the source and the specific components of exosomes. The techniques currently used include Western blots, flow cytometry, and global proteomic analysis using mass spectrometry [158]. The study of exosomes and their scientific relevance has greatly advanced, but there is a need to focus on the methods to identify the source of exosomes and especially their specific content. This will advance to the next level our research on the use of exosomes to identify the molecular basis and cellular etiology of diseases.

## Figures and Tables

**Figure 1 ijms-22-11685-f001:**
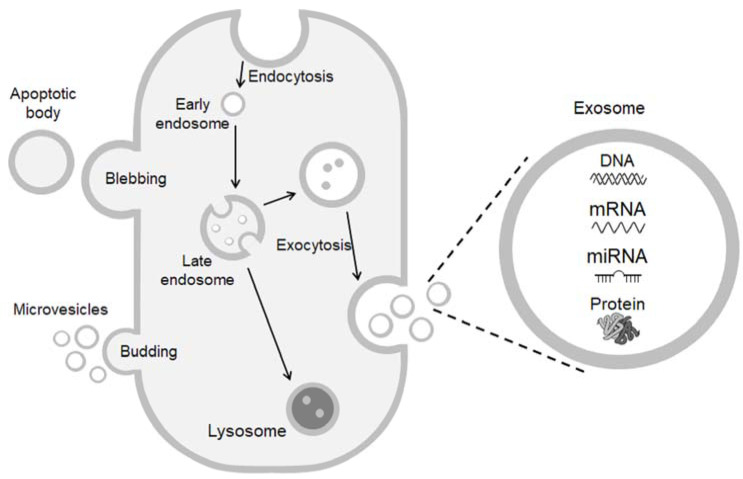
Exosomes’ formation. Exosomes containing DNA, mRNA, miRNA, and proteins are formed by the inward budding of late endosomes and the formation of small vesicles that will be released into the extracellular space through exocytosis.

**Figure 2 ijms-22-11685-f002:**
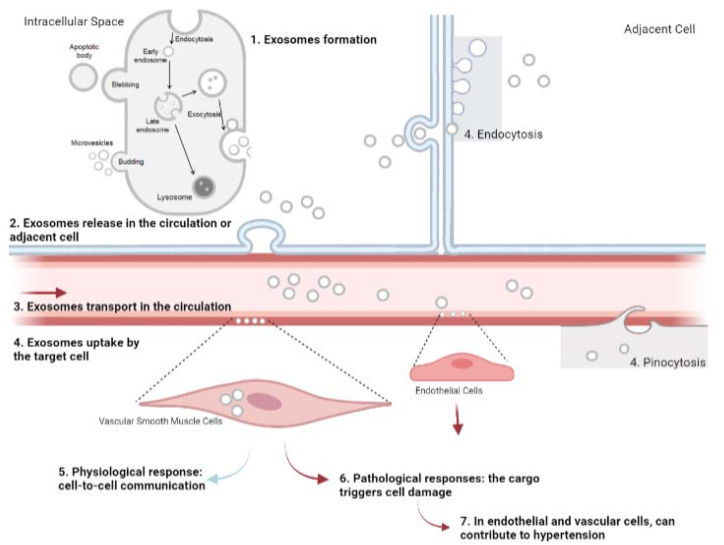
Exosomes mediate intercellular communication, contributing to hypertension. 1. Exosomes are formed by late endosomes in cells, budding into microvesicles; 2. exosomes are released by exocytosis into the circulation or in adjacent cells; 3. once exosomes are in the circulation, their cargo and affinity for the cell membrane will determine which cells are their target; 4. several types of cell, including immune cells, smooth muscle cells, and cancer cells, can uptake exosomes by different mechanisms, such as pinocytosis, phagocytosis, endocytosis, and internalization; 5. physiologically, the cargo of exosomes promotes cell-to-cell communication; 6. however, some contents of exosomes may trigger pathological responses, leading to diseases; 7. ultimately, in endothelial and vascular smooth muscle cells, the release of some miRNAs and other proteins from exosomes can cause endothelial/vascular dysfunction and contribute to the development of hypertension.

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
