# Peer review of "Exosomes as Intercellular Messengers in Hypertension"

_ijms, 2021, doi:10.3390/ijms222111685_

Round 1
Reviewer 1 Report
The manuscript entitled “Exosomes as intercellular messengers in hypertension” are impressive for me including good writing, nice organization, valuable information, attractive headline, and significance of scientific research in related fields. Despite this, I still have several concerns and hope authors address them in revised version.
- By checking plagiarism in ithenticate website, I found 54% plagiarism and it needs to be decrease in the revised version.
- For author’ address, all of authors came from same center in same institution, I don’t know the reason why look like coming from 4 different addresses?
- If searching google scholar and/or NCBI by given the key words “exosome and hypertension”, lots of published articles, but the cited references are very few.
- Does one of the authors study exosome/hypertension related projects? If so, please add your own studies in your revised version. If not, please mention your financial disclosure.
Author Response
Response to Reviewer’s Comment
Point 1: The manuscript entitled “Exosomes as intercellular messengers in hypertension” are impressive for me including good writing, nice organization, valuable information, attractive headline, and significance of scientific research in related fields. Despite this, I still have several concerns and hope authors address them in revised version.
Response: Thank you for your nice comment, we have done our best to address all your comments
Point 2: By checking plagiarism in ithenticate website, I found 54% plagiarism and it needs to be decrease in the revised version.
Response: Thank you for your pointing this out. We have edited our manuscript to decrease the % plagiarism. We do not have access to the ithenticate software, so we used Turnitin instead. We found that some words like hypertension, exosomes and preeclampsia and some sentences like ‘the mechanism of’, ‘The role of’ were highlighted. We could not change these words, but we edited the manuscript to change sentences that could be changed.
Point 3: For author’ address, all of authors came from same center in same institution, I don’t know the reason why look like coming from 4 different addresses?
Response: Thank you for your comment. The reason why the addresses looked like we are from different institutions is because we used the template provided by the journal when preparing the manuscript. This has been corrected in this version.
Point 4: If searching google scholar and/or NCBI by given the key words “exosome and hypertension”, lots of published articles, but the cited references are very few.
Response: Thank you for your pointing this out. We have included more references in the edited version of our manuscript. The following references were included: Lee et al., 2012; Aliota et al., 2016; Willis et al., 2018; Barros et al., 2017 and Bei et al., 2017.
Point 5: Does one of the authors study exosome/hypertension related projects? If so, please add your own studies in your revised version. If not, please mention your financial disclosure.
Response: Thank you for this question. All the authors study hypertension and the corresponding author submitted a grant proposal on the roles of exosomes in pregnancy-induced hypertension and is just beginning her research in this area. So, we do not have completed and published articles that we could cite.
Reviewer 2 Report
In this manuscript, Arishe et al. summarize the current knowledge about the role of exosomes as regulators and biomarkers in hypertension. This is a comprehensive and well written review that provides an interesting and brief summary on the literature related to exosomes in hypertension, which is a topic of current interest. However, some major and minor points should be addressed to improve the manuscript before publication.
Major comments
- Page 5, line 199,200. “miR-128 also methylates the VSMC gene myosin heavy chain 11 (Myh11).” As written, it appears that miR-128 itself methylates Myh11. This is incorrect. The authors should rephrase the sentence to clarify that miR-128 exerts its methylation effect via KLF4.
- Item 3.5. Exosomes in the development of preeclampsia (pages 6 and 7). The authors should emphasize the role of soluble endoglin (sEng) in preeclampsia/hypertension. Among others, sEng is increased in preeclampsia (Venkatesha et al. Nat. Med. 2006; 12: 642-9. doi: 10.1038/nm1429), and increased levels of exosomal endoglin is present in the maternal circulation of preeclamptic patients (Ermini et al. Sci. Rep. 2017; 7:12172. doi: 10.1038/s41598-017-12491-4); also, endoglin has been described in endothelium-derived microparticles from thromboembolic pulmonary hypertensive patients (Belik et al. J. Biomed. Sci. 2016; 23: 4. doi: 10.1186/s12929-016-0224-9). More importantly, sEng is not only a biomarker but also appears to play a pathogenic role in hypertension (Valbuena-Diez et al. Circulation 2012; 126: 2612. doi: 10.1161/CIRCULATIONAHA.112.101261; Peres-Roque et al. Int. J. Mol. Sci. 2020; 22:165. doi: 10.3390/ijms22010165; Gallardo-Vara et al. Cells 2020; 9:988. doi: 10.3390/cells9040988). In addition, sEng is involved in inflammation and endothelial dysfunction, both processes related to hypertension (as reviewed in Vicen et al. Cell Mol. Life Sci. 2021; 78: 2405. doi: 10.1007/s00018-020-03701-w). Please cite and comment the appropriate references.
- Figures. The manuscript contains a single figure (Figure 1) which is rather basic and appropriate for an exosome introduction. In the context of hypertension, the authors may wish to enrich the manuscript by providing an additional illustration that includes the different exosome cargos, their related functions, their altered expression levels (biomarkers) and/or their target cells. This may help the reader to visualize the overall content of this review related to hypertension.
- References. The authors have omitted some recent and relevant references:
Perez-Hernandes. Hypertension. 2021; 77: 960. doi: 10.1161/HYPERTENSIONAHA.120.16598.
Ge et al. Curr. Pharm. Biotechnol. 2021; 22: 1654. doi: 10.2174/1389201022666201231113127.
Zhang et al. Respir. Res. 2020; 21: 71. doi: 10.1186/s12931-020-1331-4.
Hogan et al. Am. J. Physiol. Lung Cell Mol. Physiol. 2019; 316: L723. doi: 10.1152/ajplung.00058.2018.
Please cite and comment as appropriate.
Minor comments
- Page 1, line 39. “….and causes of this disease, that much more crucial.” This sentence is confusing. Do the authors mean “….and causes of this disease, much more crucial.”? Please revise.
- Page 3, line 109. “exosomes” instead of “exomes”
- Page 3, line 110. “….including: 1)….” instead of “….including 1)….”. Please add colon.
- Page 4, line 146. “MicroRNA (miRNA) is a small 18-to 28- nucleotide non-coding RNA molecule. They regulate…..”. For singular/plural consistency, the authors may rephrase the sentence as follows: “MicroRNA (miRNA) are small 18-to 28- nucleotide non-coding RNA molecules. They regulate….”
- Page 4, line 150. “Argonaute” instead of “Argonuate”.
- Page 4, lines 186,187. “EGF, and NGF, REN”. Please spell out the abbreviations.
- Page 6, line 284. “…it was reported plasma exosomal miR-15a was higher…” > “…it was reported that plasma exosomal miR-15a was higher…”
- Page 6, line 290. “were” instead of “was”.
- Page 7, lines 306,307. In order to avoid repetition of “currently”, the authors may wish to modify the text as follows: “…remains unclear. Currently the only…..”
- References (pages 8-13). The numbering of the references is duplicated. Also, please follow the standard format of IJMS (journal name in italics, year in bold, volume in italics, etc.)
Reviewer 3 Report
In this review article titled “Exosomes as intercellular messengers in hypertension”, Arishe et al. describe the myriad roles exosomes play in the pathogenesis of hypertension. Hypertension is a common disease, but—as the authors explain—its etiology and pathogenesis remain incompletely understood. With increased recognition of the universal importance of exosomes in intercellular communication, a detailed review discussing the role of hypertension seems quite timely and apt. However, I felt the manuscript could be improved, as I detail below.
1, Organization: I was of the impression that the review was poorly structured. Findings from papers were overall simply mentioned in tandem without being integrated into the bigger picture. Sections/sub-sections are of very different lengths, with a great amount of detail (e.g. 3.1) in some and very little in others (4).
2, Section 4: The section is titled “Targeting exosomes in hypertension”, but very little is discussed how to target exosomes in hypertension.
3, Discussion: The discussion was very superficial and without much original insight provided by the authors. Some suggestions: “This could go a long way in developing therapeutic interventions specific to individuals”—how, specifically? “Future studies should focus on elucidating the various components of exosomes in hypertensive patients and the mechanism through which each component individually or collectively exacerbate the disease process”—how, specifically? What are some current limitations and what are the advances needed? What do the authors think is the best next step forward?
4, Figures: The manuscript only has one figure (not mentioned in the main text) concerning the biogenesis of exosomes, which is not really the main point of this review. While not strictly necessary, I feel figures depicting key messages of this review would greatly help the reader’s understanding.
5, References: Many non-trivial statements were made without a proper reference (if I reckon correctly, Int J Mol Sci does not have an upper limit to the number of references).
6, English proofreading: Overall, there were many very long sentences. Paragraph lengths were very irregular, with some abrupt transitions. I would suggest thorough proofreading to make the manuscript easier to read and to follow the logic.
Author Response
Please, see attachment

Round 2
Reviewer 2 Report
The authors have properly addressed most of my concerns. I have no further comments.
Author Response
Thank you so much for your help in making this manuscript better.
Reviewer 3 Report
In their revision, Arishe et al. address some of the concerns I have raised in my previous report, and the manuscript has improved. However, I still have concerns as I describe below.
Major comments:
1, I am still of the impression that the organization of the paper is poor, with repetitive and tangential material. Proofreading issues still remain. I furthermore still feel that the review falls short of providing a cohesive integration of and/or novel insight into the available evidence as the discussion (though somewhat improved) is still rather superficial.
2, References have been improved, but issues still remain here as well.
2-1, Lines 79-81, "The selectivity of the cell-to-cell communication...": This claim needs a suitable reference.
2-2, Lines 128-130, "Studies have shown...": References are needed.
2-3, Lines 259-261, "In physiological situations...": This claim needs a suitable reference.
2-4, Lines 309-311, "It affects...": This claim needs a suitable reference.
2-5, Lines 329-331, "Studies have shown that...": References are needed.
2-6, Lines 347-348, "Generally, it has been...": The current reference (101) is only about preeclampsia. I would suggest adding references for PH and caner as well.
3, The revised figure probably would be better as independent figures. Also, the added portion depicting the blood vessel includes too much text. I would suggest moving some of the text to the figure legend and keeping text in the figures minimal.
Minor comments:
1, Line 120: Gap junctions are not the only mode of cell communication via direct contact, so I don't think a specific referral to "gap junctions" is necessary here.
2, Line 123-4: The authors mention CA and AQP as examples of "proteins that are specific to the function of its originating cell" but do not mention the originating cell.
3, Line 224: The parenthesis before "Ang II" opens but does not close.
4, Lines 269: The authors mention "Another way", but I was not exactly sure what the first one was. Please clarify.
5, Lines 396-402: Why the quotation marks?
Author Response
Thank you so much for your help in making the manuscript better. Please see attachment.

Round 3
Reviewer 3 Report
Thank you for the revisions, changes were noted.